# Femtosecond Laser Assisted 3D Etching Using Inorganic-Organic Etchant

**DOI:** 10.3390/ma15082817

**Published:** 2022-04-12

**Authors:** Agnė Butkutė, Greta Merkininkaitė, Tomas Jurkšas, Jokūbas Stančikas, Tomas Baravykas, Rokas Vargalis, Titas Tičkūnas, Julien Bachmann, Simas Šakirzanovas, Valdas Sirutkaitis, Linas Jonušauskas

**Affiliations:** 1Femtika Ltd., Saulėtekio Ave. 15, LT-10224 Vilnius, Lithuania; greta.merkininkaite@femtika.com (G.M.); tomas.jurksas@femtika.com (T.J.); tomas.baravykas@femtika.com (T.B.); rokas.vargalis@femtika.com (R.V.); titas.tickunas@femtika.com (T.T.); 2Laser Research Center, Vilnius University, Saulėtekio Ave. 10, LT-10223 Vilnius, Lithuania; jokubas.stancikas@ff.stud.vu.lt (J.S.); valdas.sirutkaitis@ff.vu.lt (V.S.); linas.jonusauskas@ff.vu.lt (L.J.); 3Faculty of Chemistry and Geoscience, Vilnius University, Naugarduko Str. 24, LT-03225 Vilnius, Lithuania; simas.sakirzanovas@chf.vu.lt; 4Chemistry of Thin Film Materials, Department of Chemistry and Pharmacy, IZNF, Friedrich-Alexander University of Erlangen-Nürnberg, Cauerstr. 3, 91058 Erlangen, Germany; julien.bachmann@fau.de

**Keywords:** selective laser etching, 3D laser microfabrication, glass microprocessing

## Abstract

Selective laser etching (SLE) is a technique that allows the fabrication of arbitrarily shaped glass micro-objects. In this work, we show how the capabilities of this technology can be improved in terms of selectivity and etch rate by applying an etchant solution based on a Potassium Hydroxide, water, and isopropanol mixture. By varying the concentrations of these constituents, the wetting properties, as well as the chemical reaction of fused silica etching, can be changed, allowing us to achieve etching rates in modified fused silica up to 820 μm/h and selectivity up to ∼3000. This is used to produce a high aspect ratio (up to 1:1000), straight and spiral microfluidic channels which are embedded inside a volume of glass. Complex 3D glass micro-structures are also demonstrated.

## 1. Introduction

Glass micro-processing using femtosecond (fs) lasers is a vast field with capabilities of producing different structures with arbitrary geometries [1]. This can be achieved using various light-matter interaction regimes. II type modification-based selective laser etching (SLE) stands out among all of them due to the possibility of producing arbitrary shaped 3D glass structures which can be on the surface of the glass samples or embedded inside the volume of the sample. This technology was also shown to be suitable for processing crystals [2,3,4], making it even more appealing.

The underlying idea behind SLE is that the etchant etches laser-induced modified volume of dielectric substantially faster than unmodified material. To characterize it, two primary parameters can be used: etching rate, which is the speed at which modified material dissolves in the etchant, as well as selectivity, which denotes the ratio between etching rates of modified and unmodified glass. Due to different requirements dictated by various applications where SLE is used, different etching rates and selectivities might be desired. As a result, an extensive variety of works have been dedicated to understanding the underlying physical and chemical mechanisms, which might lead to different etching rates and selectivities [5]. Most of them concentrate on changing laser exposure parameters [3,5,6]. In the absolute majority of works, aqueous Hydrofluoric acid (HF) or Potassium Hydroxide (KOH) solutions are being discussed [5]. Nevertheless, varying only these parameters, it is hard to achieve selectivity which would be higher than ∼1400 [6]. This limits SLE usage as a true 3D manufacturing technique which could rival additive 3D printing [7].

This work is dedicated to expanding SLE selectivity further by employing etchant with additional organic solvents mixed in it. This technique is already used in standard lithography, which also deals with wet etching [8]. It results in different wetting and chemical peculiarities of the process, subsequently enhancing etching rate and selectivity. These capabilities are explored in this work, showing what maximum etching rate and selectivity can be achieved using this solution. Additionally, the phenomenon is explained from the chemical interaction side, allowing us to understand it towards a better exploitation in producing a high aspect ratio and/or complex embedded 3D structures.

## 2. Materials and Methods

The work was performed using a “Laser Nanofactory” (Femtika, Ltd., Vilnius, Lithuania) setup. The system is based on amplified Yb:KGW fs laser “Pharos” (Light Conversion, Ltd., Vilnius, Lithuania)). In this work, the laser parameters were set to 1030 nm wavelength, 610 kHz repetition rate, and ∼700 fs pulse duration. Positioning is performed using Aerotech linear stages and galvo-scanners, which operate in a synchronized manner to avoid any stitching-related defects. Translation velocity and average laser power used for modification inscription were −10 mm/s and 300 mW, respectively. The system also has a built-in polarization control, which allows it to be dynamically tuned during fabrication. To obtain the highest possible selectivity, linear polarization perpendicular to the scanning direction was used for selectivity testing experiments. More precise details on the setup can be found elsewhere [9]. Fused silica glass (amorphous SiO_2_) was used for the fabrication. The main idea of the experiment is to inscribe single scanning lines in the XY plane volume of the glass. This method was already presented elsewhere [10]. The process is also shown in Figure 1. Five identical lines are written in one sample to check the repeatability of the process. After the inscription process, the sample is cut in the middle perpendicularly to the written lines. The cut edge of the sample is polished and then the sample is etched in the chosen etchants. Etching was performed in an aqueous KOH solution with varying isopropanol and KOH concentrations at 85 °C. Exact concentrations will be listed where applicable. After a specified amount of time, etched channel lengths are measured by using an optical microscope. Optical microscope images were taken using Olympus IX73. By dividing channel length by etching time, the etching rate is calculated. To calculate selectivity, we needed to measure the unmodified material etching rate for each case. This was done by measuring the thickness of the bulk of glass and etching the glass bulk for 24 h. After the etching process, glass thickness was measured again and the unmodified material etching rate was calculated by dividing etched glass thickness by etching time. After all, by dividing the modified material etching rate by the unmodified material etching rate, the selectivity is calculated.

The contact angle characterization was performed by using the optical contact angle measuring and contour analysis system (KSV Instruments LTD CAM 200, The Imaging Source Europe GmbH DMK 21F04) equipped with a CCD camera (ImagingSource DMK 21F04 fps). The camera, the etchant droplet, and the illumination source equipped with a light-emitting diode were aligned into one line. Therefore, the droplet shadow was projected and captured by the digital camera. Average values of contact angle and measurement errors were obtained by software (The Imaging Source Europe GmbH DMK 21F04).

## 3. Results

We began this work by testing isopropanol’s influence on the etching rates of unmodified fused silica by changing KOH and isopropanol concentrations. The chosen range was 1–10 mol/L for KOH and 0–50% for isopropanol. The unmodified material etching rate measurement was performed by measuring the thickness of the fused silica sample before etching and then etching it for 24 h. After etching, the thickness was measured again and the etching rate of unmodified material was calculated. Since the etchant does not need to penetrate inside the volume of glass and the diffusion of the etchant is not confined, the etching rate of unmodified material should remain practically the same over the experiment time scale. Results are shown in Figure 2a. As expected, for unmodified material, the etching rate was far more influenced by the KOH concentration than the amount of isopropanol in the mixture. The peak value of ∼1.5 μm/h was achieved at the highest KOH concentration and a low 5% concentration of isopropanol.

Next, the etching rate of the modified material was measured. The experiment protocol followed the idea of inscribing horizontal lines into the material one after another. The etching rate was determined by measuring the length of a 1 h etched channel that was formed during the described process. Measured values are plotted in Figure 2. In contrast to unmodified results, the highest etching rate (∼820 μm/h) here was achieved at a moderate KOH concentration of 6 mol/L and with no isopropanol. However, another peak of the higher etching rate of ∼820 μm/h was observed at the 2 mol/L KOH concentration and with 20% of isopropanol, showing that organic solvent, under special conditions, can have a positive impact on the etching rate of laser modified material.

SLE was shown to be able to produce complex 3D glass structures, rivaling those produced using additive manufacturing out of polymers [1]. However, the complexity and size of SLE-produced structures are always limited by the selectivity, i.e., even if laser exposed regions etch much faster, unaffected material is removed as well. This results in limitations to the shape complexity, which might be detrimental to many applications, such as precise microfluidics or micromechanics. Therefore, the most important parameter of the SLE is selectivity. The selectivity calculated in this work is given in Figure 2c. The highest achieved value is ∼3000, which is more than two times higher than the highest value reported in the literature to the best of our knowledge [6]. In addition, it was achieved in a mixture containing small amounts of both KOH (2 mol/L) and isopropanol (10%). This is also far from both peak etching rates of unmodified and modified materials. If no isopropanol was used, 2 mol/L of KOH would yield a selectivity of ∼2700, which is ∼13% lower compared with the maximum value. Therefore, a positive impact on selectivity by using small amounts of isopropanol is evident. Interestingly, it is a far higher value in comparison to the selectivity of ∼500 at 10 mol/L KOH—a standard value used by multiple groups. Thus, another conclusion that can be drawn from this work is that even a reduction in KOH can help increase selectivity. However, care should be taken when applying this methodology, since a decrease in KOH concentration would result in longer etching times.

The primary idea behind introducing isopropanol to the aqueous KOH mixture is to improve the wetting properties of the liquid and, hence, allowing easier diffusion of dissolved silicate ions from the fused silica-etchant interface. Thus, to obtain further insights into acquired etching results contact angles of different ratios of KOH and isopropanol on fused silica were measured. Results are given in Figure 3. Generally, with the increase of isopropanol concentration, the contact angle becomes smaller, generally proving the idea that this organic solvent should help with wetting. However, the area with the lowest contact angle, which is between and 2 mol/L KOH and above 20% of isopropanol coincides neither with the highest etching rates nor with the highest selectivity. On the contrary, the highest etching rates are at experimental conditions with relatively high contact angles −34.5° for the highest etching rate for unmodified material and 32.1° for laser affected volume. Nevertheless, the highest selectivity value coincides pretty well with the area where the contact angle starts to drop significantly (17°) at the highest selectivity conditions. At the same time, it is not at the lowest achieved contact angle −5.2°. Therefore, some other interactions should be considered.

To understand the full picture of the process, the chemical aspect has to be considered. Indeed, when etching is performed, the resulting chemical reaction is the same in both cases with and without isopropanol. When fused quartz is exposed to a solution of KOH, potassium silicate and water are formed (see Figure 1). However, in addition to the increased wetting effect of fused quartz described above, the introduction of isopropanol into the system also has a negative influence on selectivity. The requirement for the reaction is the formation and dissolution of K_4_SiO_4_ in the reaction mixture. Potassium silicate is soluble in water (it is possible to dissolve salt with boiling water) and slightly or completely insoluble in alcohols [11]. Potassium silicate dissolves more readily under pressure at a temperature of 80 °C [12,13]. Even though the etching was carried out at a temperature of 85 °C, in a closed system, the temperature of etchants had to be lower because of the boiling point of isopropanol, which is 82.5 °C. Consequently, the higher isopropanol concentration determines the lower temperature of the mixture and the lower amount of a more suitable solvent, resulting in a slower etching process at extremely high isopropanol concentrations.
(1)$${\mathrm{SiO}}_{2(s)}+4K^{+}{\mathrm{OH}}_{\left(aq\right)}^{-}\to K_{4}^{+}{\left[{\mathrm{SiO}}_{4}\right]}_{\left(aq\right)}^{4-}+2H_{2}O_{\left(I\right)}.$$

The solubility of potassium hydroxide is ∼11 g /10 mL (28 °C) and 121 g/100 mL (25 °C) in isopropanol and water, respectively. It can be suggested that isopropanol acts as a surfactant, accumulates at the solid/liquid interface, and thereby prevents access of KOH to the surface of the silica sample. We considered a model of the behavior of isopropanol molecules at the surface during etching. The described model takes into consideration that, for a saturated concentration, a layer of isopropanol molecules is formed at the surface of the solution, rather than being diluted. The abundance of isopropanol affects the formation of a monolayer of alcohol molecules at the surface of the sample [14,15,16]. The highest selectivity value was obtained for 2 mol/L (10.89 g/100 mL, here KOH molar mass 56.1056 g/mol, isopropanol density at 28 °C is 0.7783 g/mL) KOH concentration, which corresponds to the maximum alkali solubility in isopropanol. Further increasing the alkali concentration in the mixture results in a decrease in selectivity. This evidence confirms the described model of isopropyl-alcohol monolayer formation on the surface because the etching reaction takes place only at the surface and the etchant contact. However, to prove this hypothesis, further studies need to be performed.

Thus, overall, while a lower contact angle helps with easier etchant and product diffusion between the solution and the fused silica interface, behaviors of isopropanol, such as lower boiling point and limited dissolution of KOH, result in a slowed down reaction, forming an ideal balance at experimental conditions of 2 mol/L KOH and 10% isopropanol.

Despite an impressive etching rate together with selectivities which were demonstrated, etching rates tend to diminish. We have measured how the etching rate is changing over time. The results of the etching rate after 2, 4, 8 and 12 h are depicted in Figure 4a–d. Comparing etching rates after the mixture was just mixed after 2 additional hours’ etching rates drop approximately twice (a). After 4 h, etching rates decrease twice again. Finally, after 8 and 12 h, the etching rate is between 100 and 200 μm. The explanation for such a decrease in etching rate is simple—the deeper the etchant needs to penetrate into the channels, the more it dilutes and struggles to etch the channel at the deepest point. Nonetheless, here we can notice the special property of the etchant with isopropanol. Isopropanol allows the maintenance of a higher etching rate. Possibly, isopropanol induces more effective penetration into the channel. Thus, another positive side of isopropanol is the possibility to not only achieve but also maintain a higher etching rate for longer. In general, according to these graphs, we can conclude that if high selectivity and a high etching rate are desired, the etching time should be short enough (shorter than 2–3 h). Otherwise, it is impossible to maintain selectivity and an aspect ratio higher than 1000, because of the loss of etching rate during longer etching times. This might be problematic if relatively big or long structures are needed.

After we performed some experiments we have noticed one more interesting property which is affected by the etchant. We have observed that structures etched with lower KOH concentration etchants tend to have worse surface quality than structures etched with high concentration etchants. To test this property we inscribed the surface in the XY plane and etched it away with different concentrations of etchant. After all, surfaces were inspected with a Scanning Electron Microscope (SEM). SEM pictures of made surfaces are shown in Figure 5a–g. Although surfaces look different, surface roughness was measured for these surfaces by using an optical profilometer. The correlation between surface roughness and etchant concentrations is shown in Figure 5h. The higher the etchant concentration is, the lower the surface roughness that could be obtained. On the other hand, looking back to previous results, lower concentrations of etchant allow obtaining higher sensitivities. For comparison in Figure 5h, we added a surface etched with 5% HF solution with a surface roughness of around 200 nm root mean square (RMS). Usually, HF shows substantially lower selectivities than KOH [5]. Thus, here we conclude that the higher the selectivity, the higher the surface roughness obtained. However, in most applications, it is a desire to maintain both. To get that result, combined etching techniques need to be used. For example, etch the structure with low concentration KOH etchant and smooth the surface with low concentration HF solution. The hybrid etching technique was already demonstrated elsewhere [17], on the other side, reasons are different, a combination of a few different types of etchant, such as HF and KOH, allow us to obtain the highest throughput because HF shows a higher etching rate for effective large volume removal, meanwhile KOH allows us to achieve high selectivity for precise feature etching.

From a practical perspective, the best way to demonstrate the potency of the developed methodology was the formation of high aspect ratio embedded fluidic channels. Indeed, while SLE shows huge promise in microfluidic system fabrication, in many cases channels have to be produced open and then be sealed afterwards using other techniques, such as fs laser welding [18], or have limitations on their physical dimensions. Here we demonstrate high aspect ratio channels embedded in the glass and etched out using the highest, KOH enabled selectivity. The thickness of the sample is 2 mm, while the channels are ∼10 μm in diameter. This places their aspect ratio at 1:1000. Channels were filled with liquid and optical images were taken from the side (Figure 6a,b). The presented optical microscope pictures allow us to estimate that the diameter is well maintained through the channel while the surface roughness is around hundreds of nanometers RMS. It is important to note that, due to the thickness of a sample, an objective with automated aberration correction might be required if this methodology is used with relatively thick samples (more than 1 mm). In the future, Spatial Light Modulators (SLM) can be used for the same purpose with more parameter control [19]. Non-diffracting Bessel beam-based fabrication might also be attractive to fabricate such long and straight channels with a very high aspect ratio [20].

The primary attraction of SLE is the possibility to produce complex embedded glass structures, such as microfluidic channels. While the straight channels shown so far might be used for some application, the possibility to have bent channels opens a lot of new possibilities for applications. One of them is passive particle separation. Indeed, spiral channels allow for separating particles due to their size without using any additional elements [21], such as filters [18]. The spiral-based separator provides a very high throughput, as none of the channel cross-section is blocked by the filter. To demonstrate that such a channel is possible, a prototype of an embedded spiral channel system was made. The combined length of the channel is 7 mm, with a channel width of 100 μm, resulting in an effective aspect ratio of 1:70. Testing with water showed nonrestrictive flow, which means that the channel is well etched and has no leftovers (Figure 6b). Due to the nature of the SLE, the inlet and outlets of such a system can be produced during the same fabrication step, simplifying production.

Another exemplary structure demonstrated in this work is the 3D fullerene model. An SEM picture of this structure is demonstrated in Figure 7 This structure is also a high aspect ratio and porosity complex 3D structure example. By using a combined etching technique, we achieve both high surface roughness and high selectivity which creates a possibility for complex architecture high aspect ratio structure production. For the first step of etching, 2 mol/L KOH solution with 10% of isopropanol is used to maintain the highest selectivity. Subsequently, the structure is etched additionally in 5% HF solution to smooth the surface. A fullerene structure demonstration proves that SLE is a suitable technology for a high aspect ratio porous complex structures fabrication which could lead to glass metamaterials fabrication.

## 4. Discussion

Three dimensional (3D) manufacturing, especially at scales down to the µm range, is becoming increasingly explored by academia, with industry following closely behind. So far it has been considered that 3D additive manufacturing is the preferred option due to its flexibility and wide array of possibilities to realize it [7,22]. However, while there are many technologies for it [23], on this scale, fabrication out of inorganic compounds is difficult. If direct fs fabrication is considered, special hybrid materials and/or nanocomposites need to be used [24,25,26,27], with potentially limited fabrication windows and multiple post-processing steps needed to realize it. SLE would seem to be an attractive candidate to potentially substitute complicated additive methodologies if 3D glass microstructures are needed. Nevertheless, the main problem with SLE was always selectivity, limiting how complex 3D structures can be [1]. Results shown in this work show the potential of SLE to have good enough selectivity to be directly compared to additive manufacturing. The SLE process also involves fewer technological steps than, let us say, glass manufacturing using optical 3D printing [28,29]. At the same time, SLE requires the usage of potentially hazardous chemicals (HF and KOH) which can also limit its proliferation. Therefore, while SLE is approaching additive manufacturing in terms of possibilities to produce 3D microstructures, there are some distinct advantages and disadvantages involved with this technique.

Another important takeaway from this work is the dynamics of surface roughness depending on the etching parameters. Indeed, there are many technologies tested for the production of micro-optical elements [30]. For this discussion, we will again compare SLE with additive fs-based printing, where it is inherently easy to achieve surface roughness suitable for optics manufacturing (bellow 10 nm RMS) [24]. As a result, there is a multitude of works using additive fs-based manufacturing for complex micro-optics production [31,32,33]. However, additive manufacturing forces the usage of organic materials for fabrication, which inherently have a relatively low laser-induced damage threshold [34]. Attempts to manufacture inorganic micro-optics using additive manufacturing exist, but these involve all the previously listed challenges, plus require the performance of all the manufacturing steps with extreme attention to deformations that occur during such processing [35]. Direct micro-optic manufacturing using subtractive processing is limited, as ablation and SLE do not have good enough surface roughness for optics. This leads to the necessity to use annealing [36] or polishing [37] after laser exposure, making shape control quite hard. However, in this work, we show that by manipulating the etchant, both a complex 3D shape and relatively good surface roughness can be achieved. While it is still not of optical quality, it points to the possibility of using etchant manipulations for single-step surface smoothing during etching itself. This is an area of research worth exploring in the future.

## 5. Conclusions

Overall, this work demonstrated the general dynamics of etching rates as well as selectivity when the concentrations of KOH and isopropanol are varied in the etchant. The highest etch rate of unmodified material is ∼1.5 μm/h, which is achieved with the highest tested concentration of KOH (10 mol/L) and a minor amount of isopropanol (5%). This is in sharp contrast to the highest etch rate of laser effected volume (∼820 μm/h), which is acquired at 6% KOH and no isopropanol. Finally, the highest achieved selectivity is ∼3000, which is around two times higher than the highest selectivity reported in the literature to the best of our knowledge. It is achieved at a minor concentration of KOH and a small amount of isopropanol. This result can be explained by a minimized etching rate of unmodified material and a high etch rate of modified glass. Both of these effects are a result of changes in the wetting properties of the etchant alongside the modified diffusion properties of the formed chemical constitutes of the reaction. On the other hand, over time, the etching rate drops down significantly to 100–200 μm for all etchants. Thus, to maintain a high etching rate, the etching process needs to be quite short—up to 4 h. In this work, we have also shown that even the surface roughness of the etched sample depends on the etchant type and its concentration. Low concentration etchants make rough surfaces, meanwhile, high concentration KOH or HF etchants make smoother surfaces. Hence, to maintain both high selectivity and surface quality, combined etching techniques need to be used. To show how all of these improvements can be exploited for various 3D structures, such as the model of fullerene molecule, embedded curves and straight channel systems in the glass were fabricated. The demonstrated straight channels aspect ratio is in the range of 1:1000 and spiral channel aspect ratio is in the 1:70 range with good repeatability of the fabrication. Fluidic testing showed no left-overs or debris from the channel point to well-etched channels. Complex 3D structures were also made, showing that with this improvement 3D SLE is getting ever closer to the capabilities of additive micro-and nanomanufacturing. Therefore, the results of this work allow SLE to become even more suitable for high-precision 3D glass structure fabrication.

## Figures and Tables

**Figure 1 materials-15-02817-f001:**
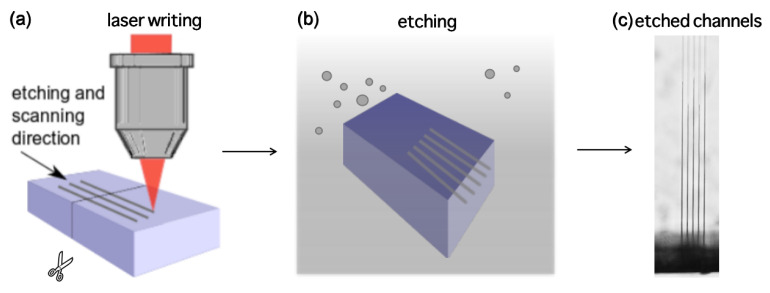
The sequence of the performed experiment. (**a**) First of all, laser modifications are inscribed inside the volume of glass in the XY plane. The sample is cut in the middle perpendicularly to the inscribed lines, the cut edge is polished. (**b**) Then, etching is performed in various enchants. (**c**) Finally, etched single line channel lengths are measured under the optical microscope.

**Figure 2 materials-15-02817-f002:**
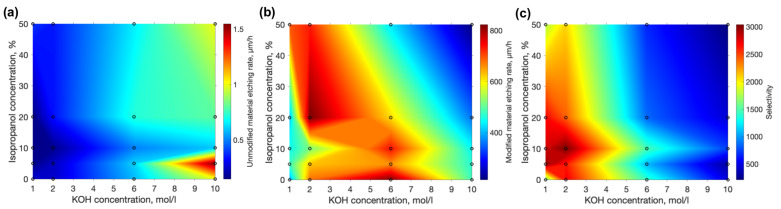
Etching rates of unmodified (**a**) and laser modified (**b**) fused silica acquired with different KOH and isopropanol concentrations. The highest achieved values are ∼1.50 μm/h and ∼820 μm/h respectively. (**c**) selectivity is derived by dividing results of part (**b**) by part (**a**). The highest value is ∼3000 at 2 mol/L KOH and 10% isopropanol, showing a clear tendency that lower KOH concentrations and a moderate amount of isopropanol positively impact the selectivity.

**Figure 3 materials-15-02817-f003:**
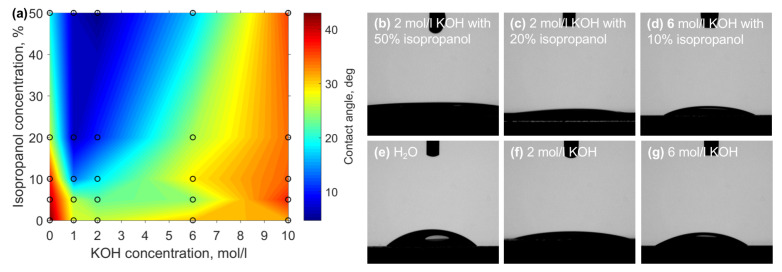
(**a**) Contact angle at different mixtures of KOH and isopropanol. While at the point of highest selectivity contact angle is relatively reduced, it is not the lowest measured value. (**b**–**g**) different etching solutions contact angle measurement pictures.

**Figure 4 materials-15-02817-f004:**
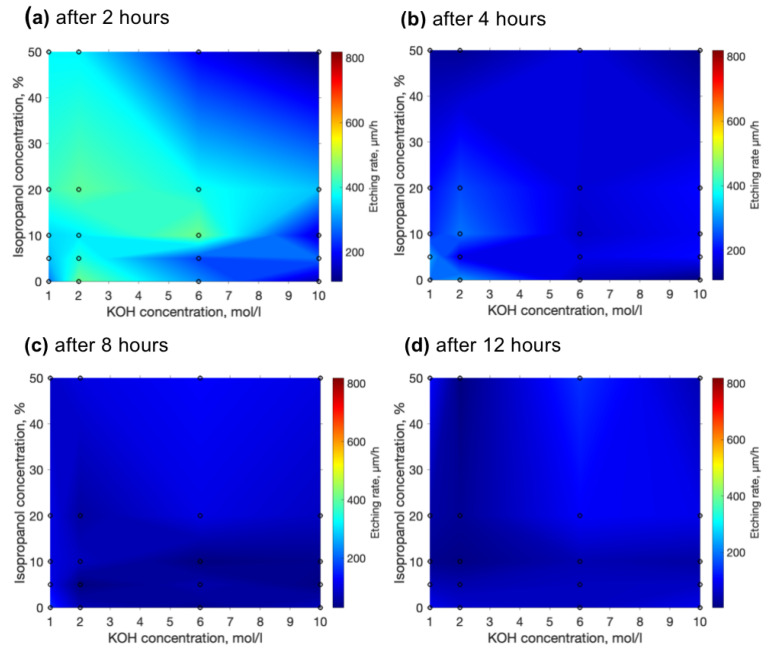
Etching rate evolution over time. (**a**) etching rate after 2 h of etching, (**b**) after 4 h, (**c**) after 8 h, and (**d**) after 12 h.

**Figure 5 materials-15-02817-f005:**
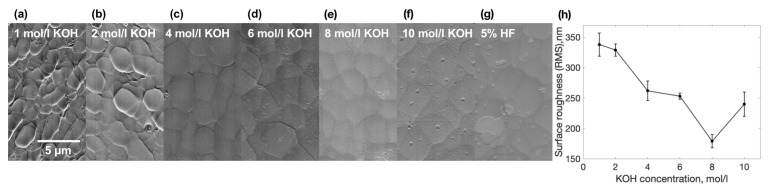
(**a**–**g**) SEM micrographs of surfaces etched in various concentrations etchants. (**h**) Surface roughness dependency on etchant concentration.

**Figure 6 materials-15-02817-f006:**
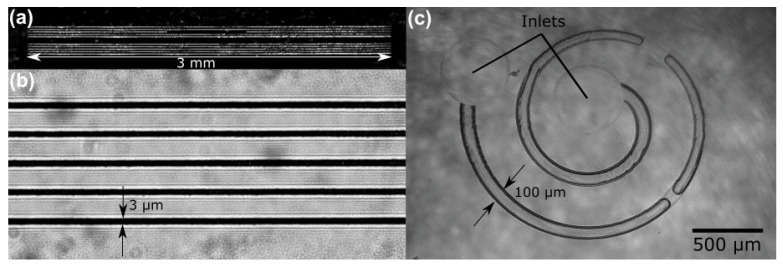
(**a**,**b**) Enhanced contrast pictures of straight-through high aspect ratio (1:1000) channels embedded inside fused silica. (**c**) microfluidic system within the volume of glass, consisting of two outlets and a spiral channel. All of the structures are etched using the best KOH and isopropanol ratio solution (2 mol/L and 10% appropriately).

**Figure 7 materials-15-02817-f007:**
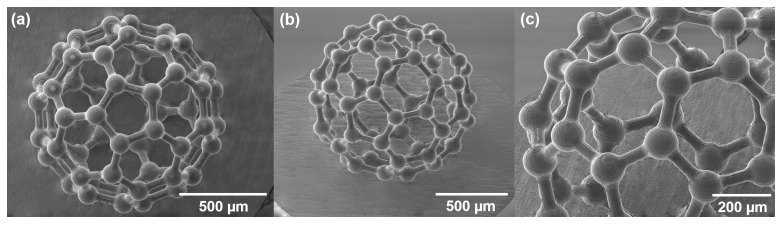
Model of fullerene molecule made out of glass. (**a**) top view of the fullerene structure, (**b**) isometric view to the whole structure, and (**c**) closed look to the same structure. This structure represents the possibility of a high aspect ratio, high surface quality, and high porosity structure which is similar to metamaterials.

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
