# Peer review of "Femtosecond Laser Assisted 3D Etching Using Inorganic-Organic Etchant"

_materials, 2022, doi:10.3390/ma15082817_

Round 1

Reviewer 1 Report

  This manuscript reprisents how the capabilities of selective laser etching technology can be improved in terms of selectivity and etch rate by applying an etchant solution based on KOH, water, and isopropanol mixture. By varying the concentrations of these constituents the wetting properties, as well as chemical reaction of fused silica etching, can be changed, allowing to achieve etching rates in modified fused silica up to 800 µm/h and selectivity up to ∼3000. This is used to produce a high aspect ratio (up to 1:1000) straight and spiral microfluidic channels which are embedded inside a volume of glass. Also, complex 3D glass micro-structures are demonstrated.
  The manuscript is well organized and will attracts many researchers in this field. Especially, the demonstration of 3D micro-fulleren structure is splendid.
  For these reasons, I recommend that this manuscript will be published as it is.

Reviewer 2 Report

This manuscript proposed a method to improving the selectivity and etching rate of SLE by adding isopropanol to the traditional etch solution. Through exploring the parameters of concentration of etchant component, the author characterized the influence of KOH and isopropanol concentration on etching rate and selectivity. In summary, the manuscript is interesting, however, it needs further improvement. This reviewer has a few major comments as follows:

  1. More supporting information about the experimental data are needed in this manuscript. The authors just showed the final summary results in Figure1 and Figure3, however, the corresponding photos of the samples were missing.
  2. It is necessary to add a schematic diagram of the method which are used to characterize the etching rate, although the authors has already cited other's experimental protocol in line 72.
  3. The surface roughness part have little contribution to improving the SLE selectivity and etching rate, although it may be used for 3D glass manufacturing.
  4. The section of displaying model of fullerene molecule seems irrelevant to the main theme in the manuscript.
  5. The authors are suggested to make the manuscript be more readable by not using long sentences. For example, line 48 can change into ‘In order to get the highest possible selectivity, linear polarization perpendicular to the scanning direction was used for selectivity testing experiments’, and so on.
  6. There are many abbreviations in the manuscript. The first time you use an abbreviation in the manuscript, both the spelled-out version and the short form should be presented, such as the abbreviations RMS in line 177 and SLM in line 200.

Reviewer 3 Report

The key achievement described in this article seems like the far enhanced selectivity derived by a combination of KOH and isopropanol. Authors specified the maximum selectivity of ~3000, which was calculated using the value obtained from etching of unmodified glass for 24 hours. However, as for the modified glass, etching was done for 1hr. Since the etch rate deteriorates over time, as authors mentioned as well, the comparison needs to be made with two glasses equally treated. As such, the selectivity appears to be overestimated. This issue should be taken care of. 

Another comment less critical is a typo error in Eq. 1; [SiO4]4+ should be [SiO4]4-. 

Round 2

Reviewer 2 Report

The authors have addressed almost all of my concerns. To fabricate micro/nano structures using wet etching method, the authors are suggested to cite below literatures:
International Journal of Extreme Manufacturing, 2021, 3(3): 35104;
Adv. Funct. Mater. 2021, 31(32), 2103298